

# Ecological niche divergence or ecological niche partitioning in a widespread Neotropical bird lineage

Jacob C. Cooper

Department of Biology, University of Nebraska at Kearney, Kearney, NE, United States of America
Biodiversity Institute, University of Kansas, Lawrence, KS, United States of America
Negaunee Integrative Research Center, Field Museum, Chicago, IL, United States of America

## ABSTRACT

Ecological niche divergence is generally considered to be a facet of evolution that may accompany geographic isolation and diversification in allopatry, contributing to species' evolutionary distinctiveness through time. The null expectation for any two diverging species in geographic isolation is that of niche conservatism, wherein populations do not rapidly shift to or adapt to novel environments. Here, I test ecological niche divergence for a widespread, pan-American lineage, the avian genus of martins (*Progne*). The genus *Progne* includes migrant and resident species, as well as geographically restricted taxa and widespread, intercontinentally distributed taxa, thus providing an ideal group in which to study the nature of niche divergence within a broad geographic mosaic. I obtained distributional information for the genus from publicly available databases and created ecological niche models for each species to create pairwise comparisons of environmental space. I combined these data with the most up-to-date phylogeny of *Progne* currently available to examine the patterns of niche evolution within the genus. I found limited evidence for niche divergence across the breeding distributions of *Progne*, and much stronger support for niche conservatism with patterns of niche partitioning. The ancestral *Progne* had a relatively broad ecological niche, like extant basal *Progne* lineages, and several geographically localized descendant species occupy only portions of this larger ancestral niche. I recovered strong evidence of breeding niche divergence for four of 36 taxon pairs but only one of these divergent pairs involved two widespread species (Southern Martin *P. elegans vs.* Gray-breasted Martin *P. chalybea*). Potential niche expansion from the ancestral species was observed in the most wide-ranging present-day species, namely the North American Purple Martin *P. subis* and *P. chalybea*. I analyzed populations of *P. subis* separately, as a microcosm of *Progne* evolution, and again found only limited evidence of niche divergence. This study adds to the mounting evidence for niche conservatism as a dominant feature of diversifying lineages, and sheds light on the ways in which apparently divergent niches may arise through allopatry while not involving any true niche shifts through evolutionary time. Even taxa that appear unique in terms of habitat or behavior may not be diversifying with respect to their ecological niches, but merely partitioning ancestral niches among descendant taxa.

Corresponding author
Jacob C. Cooper, cooperj2@unk.edu

[1]Portions of this text were previously published as part of a preprint (*Cooper, 2024*).

## INTRODUCTION

Species' ecological niches, like morphological traits, evolve *via* natural selection, thus altering their ecological niches and geographic potential through time (*Engler et al., 2021*)[1]. These processes are particularly important for diversification among closely related species, as niche divergence and diversification can lead to or reinforce speciation (*Hu et al., 2015*; *Cuervo et al., 2021*; *Şahin et al., 2021*). Such ecological shifts can lead to dramatic shifts in geographic distributional potential, and may follow predictable patterns through evolutionary time (*Cobos et al., 2021*). For example, insular species often shift from lowland to montane situations through time (*Ricklefs & Cox, 1972*; *Kennedy et al., 2022*), whereas the opposite tendency (highlands to lowlands) may dominate in continental settings (*van Els et al., 2021*).

Despite frequent opportunities for species to adapt and evolve with respect to their ecological niches, ecological niche conservatism appears to be the norm within most species (*Peterson, Soberón & Sanchez-Cordero, 1999*; *Peterson, 2011*; *Khaliq et al., 2015*; *García-Navas & Westerman, 2018*), at least with respect to coarse-resolution environmental conditions (*Comte, Cucherousset & Olden, 2017*). Such conservatism has been argued to be a contributing factor in diversification dynamics, especially in systems in which conserved niches through time force populations into allopatry and allow independent evolution to occur (*Prigogine, 1987*; *Vrba, 1993*; *Kozak & Wiens, 2006*). Indeed, in birds, secondary contact is often identified as a driving force for character divergence, including in ecological niches (*Endler, 1977*; *Seddon & Tobias, 2007*; *McCormack, Zellmer & Knowles, 2009*).

Three major scenarios for niche evolution are thus available for allopatric and parapatric populations sharing a recent common ancestor: (1) descendant populations become wholly allopatric and undergo no appreciable niche differentiation; (2) descendant populations occupy different parts of their ancestor's ecological niche in allopatry or parapatry, adapting to these specific conditions and partitioning ecological space upon subsequent secondary contact; and (3) one or more descendant populations are able to adapt to new environments and occupy novel ecological niches (Fig. 1). Scenario 1 appears to be the norm, but these patterns may be overridden in deeper time by the open exchange of genes between populations after secondary contact, as has occurred with raven lineages (*Corvus* spp.) in North America (*Omland, Baker & Peters, 2006*). Scenario 2 is perhaps best exemplified by *Poecile* chickadees within North America: two lineages (Carolina Chickadee *P. carolinensis* and Black-capped Chickadee *P. atricapillus*) have a narrow but distinct hybrid zone within which each species can survive, with the socially dominant *P. carolinensis* slowly pushing northwards as the climate warms (*Mostrom, Curry & Lohr, 2020*). These situations can also lead to complicated hybrid zones, as in taxa that now exist in secondary contact after retreating to different Pleistocene refugia (*e.g.,* members of the Yellow-rumped Warbler *Setophaga coronata* complex) (*Hubbard, 1969*; *Milá, Smith & Wayne, 2007*). Scenario 3 is perhaps less frequent, but is likely manifested within genera such as *Baeolophus* titmice, in which the interior western Juniper Titmouse *Baeolophus ridgwayi* exists in drier, more xeric conditions than all of its congeners (*Cicero, Pyle & Patten, 2020*).
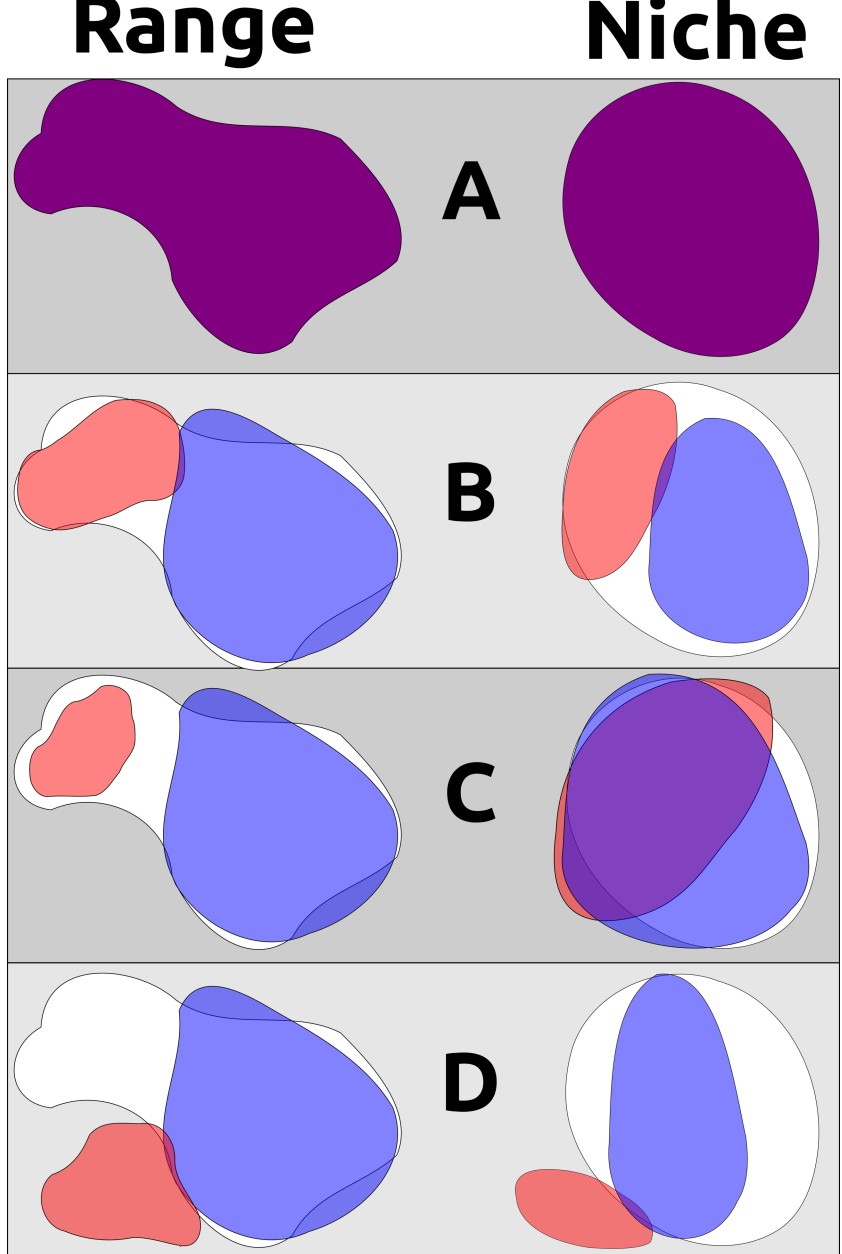

**Figure 1** **A non-exhaustive demonstration of different ways geographic range and ecological niche can evolve independently.** (A) An example ancestral species' range and ecological niche, with outlines of these characters shown in the background for subsequent descendant examples; (B) two descendant populations partition the ancestor's geographic range and ecological niche, such as when a pan-archipelagic species diversifies within individual island groups into discrete ranges and niches that are different from each other but not from the ancestral species (*e.g.*, *P. cryptoleuca* and *P. dominicensis*); (C) two descendant populations occurring in allopatry but occupying non-differentiable ecological niches and exhibiting niche conservatism; (D) two descendant populations where one occupies a portion of the ancestral range and ancestral niche while the other has evolved 

**Figure 1 (…continued)**
to occupy a new range and new niche (assuming that the ancestral species was occupying the entirety of its potential geographic distribution and the entirety of its ecological niche). Note that B and C are indicative of niche conservatism through time, even though pairwise tests of B may show ecological niche divergence in the present day.

One genus that likely has undergone multiple modes of diversification is the martin genus *Progne* (Aves: Hirundinidae). Consisting of nine species, three of which are polytypic, these aerial insectivores are distributed from central Canada to southern Argentina, nesting in cavities in trees, rocks, man-made structures, and even occasionally in the ground (*Allen & Nice, 1952*; *Pistorius, 1975*). Northern and southern representatives of the genus are migratory, with several resident taxa overlapping with these species in the Tropics during non-breeding periods (*Billerman et al., 2020*). Movements of *Progne* species appear to be complex, with some species undergoing seasonal inter-regional movements between primary and secondary wintering areas thousands of kilometers apart (*Siddiqui, 2017*). Even daily movements of *Progne* can cover large distances, with coastal, sea-level-nesting Peruvian Martins *P. murphyi* reaching elevations as high as 1,500 m (*Parker III, Stotz & Fitzpatrick, 1996*; *Luo, 2020*) and Purple Martins *P. subis* foraging as high as 1,889 m above the ground (*Helms et al., 2016*), sometimes quite far from nest sites (*Corman, 2005*). The combination of aerial feeding habits and migratory behaviors enables *Progne* to cover long distances as migrants and as vagrants, as demonstrated by records *P. subis*, a species that breeds broadly across central North America and winters in central South America, from Alaska, Ireland, Scotland, the Azores, and the Falklands (Malvinas) (*eBird, 2012*; *Quigley, 2018*; *Brown, Airola & Tarof, 2021*).

Martins show high levels of phenotypic conservatism, such that winter distributions for many species are unknown or are only being confirmed in recent years; many migrant or away-from-breeding-range individuals (especially among the 'white-breasted' martin complex of the Caribbean and Mexico) are not identifiable in the field (*Fang & Schulenberg, 2020*; *García-Lau & Turner, 2021*; *Perlut & Williams, 2021*). Distributions vary greatly in extent: the Galápagos Martin *P. modesta* (*Roper, 2020*) and *P. murphyi* (*Luo, 2020*) inhabit geographically limited xeric coastal areas whereas most widespread species are found across continents either seasonally or year round. An extreme example is the Gray-breasted Martin *P. chalybea*, which breeds across the entire latitudinal breadth of the Tropics from Mexico to Argentina (*Lagasse, 2020*).

The genus *Progne* was thus well suited for exploring the nature of niche evolution and determining whether species in a large geographic radiation exhibit patterns of ecological niche divergence or patterns of ecological niche partitioning consistent with the null hypothesis of ecological niche conservatism. Given that this genus is monophyletic and its phylogenetic relationships have been well documented (*Moyle et al., 2008*; *Brown, 2019*), *Progne* are well-suited to exploring the evolutionary history of ecological niche traits. *Progne* martin species occur in allopatry, parapatry, and sympatry in their breeding distributions, allowing for tests of niche conservatism between pairs of species that are of varying relatedness and that possess varying levels of geographic overlap. Reconstructing

niche evolution in this genus will also shed light on what kinds of evolutionary shifts lead to local endemism and will provide a continental comparison to island speciation cycles.

## METHODS

### Occurrence data

The genus *Progne* was selected for this study, given its broad range throughout North and South America and its confirmed monophyly (Table 1) (*Brown, 2019*). Occurrence data for all *Progne* martin species were downloaded from the Global Biodiversity Information Facility (GBIF) on 21 Feb 2022 (Global Biodiversity Information (*Global Biodiversity Information Facility, 2022*).

Data were processed using R 4.1.2, 4.2.0, and 4.2.1 (*R Core Team, 2022*), relying on the general data packages of *tidyverse* (*Wickham et al., 2019*) and *data.table* (*Dowle & Srinivasan, 2019*), and the general spatial packages *maptools* (*Bivand & Lewin-Koh, 2022*) , *raster* (*Hijmans, 2022*), *rgdal* (*Bivand, Keitt & Rowlingson, 2022*), and *sf* (*Pebesma, 2018*). Data were reduced to presence-absence data based at localities, with duplicate sightings of species from single sites removed and localities with an uncertainty of >10 km removed. The threshold of 10 km was selected to ensure that records are fairly close to the actual site of observation, and to reflect the "best practices" encouraged by the eBird database, one of the largest data contributors to the GBIF database, which encourages users to keep lists at or below 8 km (rounded up to 10 for the purposes of this study) (*eBird, 2012*). Distributions of individual species were superimposed on country borders using *rnaturalearth* (*South, 2017*) and *rnaturalearthhires* (*South, 2022*), and occurrences of each species were compared to known distributions for each taxon (*Billerman et al., 2020*). Outlier occurrences for each taxon were removed or re-identified depending on their location, the taxonomic history for the species in question, and on the species' residency status in a given region (detailed in Supplemental Materials; 01_data_format: Removing Outliers). These steps are necessary given the number of misidentifications and low-confidence identifications in the database at the time of download and given the long-distance dispersal ability of *Progne*. Based on published distributions and breeding records, I created dispersal areas by hand based on major biogeographic barriers (*e.g.*, straights, crests of mountains) referred to as **M**s (*Soberón & Peterson, 2005*; *Cooper et al., 2021*). In addition to being used in helping restrict models to accessible biogeographic areas, these models were used to help identify and filter erroneous observations for each *Progne* taxon.

Once points were concatenated, I ran a custom 'rarefy' code provided by Dr. J. D. Manthey (Texas Tech University) to thin occurrences using the *R* package *fossil* (*Vavrek, 2011*). Records were thinned to ensure that points were spatially independent (reducing bias in certain environments) and to reduce the computational load for geographically widespread, frequently reported species. Most species were thinned with a threshold of 20 km, twice the restriction used for data uncertainty, to reduce the likelihood of records being from the same general site and to account for large number of sightings for many species such as *P. subis*. Extremely localized species, namely *P. modesta* and *P. murphyi*, were thinned to a 10 km threshold (the same as the uncertainty buffer). This thinning step

**Table 1 Taxa used in this study.** Part (A): a list of all Progne species with subspecific breakdown for all taxa except subis. Part (B): distributions of study populations within *P. subis* used in this article and their respective subspecific groups.

| English Name | Scientific Name | Distribution |
|---|---|---|
| **(A)** | | |
| Purple Martin | *Progne subis* | Three subspecies (B). Broadly across North America; winters in C South America. |
| Cuban Martin | *Progne cryptoleuca* | Monotypic. Breeds in Cuba; winters in C and E Brazil. |
| Caribbean Martin | *Progne dominicensis* | Monotypic. Breeds throughout the Caribbean except Cuba; winters C and E Brazil. |
| Sinaloa Martin | *Progne sinaloae* | Monotypic. Breeds in Sierra Madre Occidental, Mexico; winter range uncertain. |
| Gray-breasted Martin | *Progne chalybea* | Western Mexico (*warneri*); Widespread in lowlands from Mexico to Brazil (*chalybea*) and from Brazil to Argentina (*macrorhamphus*). Far N and S populations migratory. |
| Southern Martin | *Progne elegans* | Breeds primarily in Argentina, becoming scarce as far north as Bolivia. Winters in C South America. |
| Peruvian Martin | *Progne murphyi* | Desert coasts of Peru and far N Chile. Possibly a local migrant within range. |
| Galápagos Martin | *Progne modesta* | Galápagos Islands. |
| Brown-chested Martin | *Progne tapera* | Largely resident throughout humid tropical South America (*tapera*); migrant from S South America to N and C South America (*fusca*). |

| Scientific Name | Breeding Distribution |
|---|---|
| **(B)** | |
| *Progne subis subis* | Alberta east to Nova Scotia, southwards across plains/woodlands to N Tamaulipas and S Florida. Nests primarily in artificial cavities; will nest in natural cavities. |
| *Progne subis arboricola* (Interior) | S Idaho and N Colorado south through southern Arizona and New Mexico. Southern limits uncertain; parapatric with *hesperia*. |
| *Progne subis arboricola* (Pacific) | British Columbia S to N Baja California Norte, primarily E of Cascades & Sierra Nevada. Artificial cavities in N (where intergrades with *subis*, natural cavities in S) (*Baker et al., 2008*). |
| *Progne subis hesperia* | Primarily confined to Sonoran Desert, from C Arizona southwestwards to Sonora, Baja California Sur. Nests primarily in natural cavities, especially in columnar cacti. |
| *Progne* cf. *subis* (Unknown Mexican pops.) | Possibly S Arizona & S New Mexico southwards through Michoacan & Mexico (state). Taxonomic status confusing, but often ascribed to *subis* (*Brown, Airola & Tarof, 2021*). |

also served to remove repeated observations from the same general location for species that are frequently sought at the same known breeding locations, such as *P. sinaloae* along the Durango Highway in Mexico (*Lethaby & King, 2010*). This last step was particularly important near large population centers or known colonies for rarer taxa, where records can be much denser than rural areas, potentially introducing biases in the models.
Occurrence data were largely restricted to April–July (Northern Hemisphere) and October–January (Southern Hemisphere) for migratory taxa to compare breeding niches, with minor adjustments for individual species based on their migratory patterns (see Supplemental Material; 01_data_format: Limit Dates). I focused on summer distributions as winter distributions are less well-known and are data-depauperate, especially for long-distance migrants. These restrictions ensured that comparisons of taxa encompassed similar phenological periods, and that records could be identified with greater confidence, as species' ranges are known to be largely discrete in breeding season. Non-breeding distributions of many migratory taxa remain incompletely known, such that whether non-breeding distributions of several species pairs are wholly sympatric is unknown (*Perlut, Klak & Rakhimberdiev, 2017*; *Fang & Schulenberg, 2020*; *Turner, 2020a*; *Brown, Airola & Tarof, 2021*; *García-Lau et al., 2021*; *García-Lau & Turner, 2021*; *Perlut & Williams, 2021*).

These steps were also applied to subspecies of *Progne subis* for analyses within a polytypic migratory taxon. I did not repeat these analyses with Brown-chested Martin *P. tapera*, the other polytypic *Progne* with migratory populations, as the distributions of migratory and non-migratory populations do not appear to be discrete and many records within the GBIF database are not identified to subspecies (*Turner, 2020b*). For *Progne subis*, three subspecies are described, but the limits of their distributions are not well defined (*Brown, Airola & Tarof, 2021*). Specifically, among western populations, lines of evidence for subspecies assignment, behaviors, and nesting preferences vary greatly across the species' range. As such, I subdivided *P. subis* into the following populations for analysis: nominate *P. s. subis* of eastern North America; core *P. s. arboricola* in the Rocky Mountains as far south as the Mogollon Rim; *P. s. hesperia* of the Sonoran desert; Pacific coast *P. s. arboricola* of California north to British Columbia; and interior Mexican populations of unknown taxonomic status from the mountains south of the Mogollon Rim to southwestern Mexico (*Brown, Airola & Tarof, 2021*).

## Environmental data

Environmental data were drawn from the ENVIREM dataset (*Title & Bemmels, 2018*). I removed count-format data restricting the data to continuous, raster-format variables representing terrestrial conditions. I retained elevation in the analyses but I removed the terrain roughness index, as elevation is known to affect the physiology of birds (and thus may affect nest site selection) but general terrain roughness is likely to affect *Progne* only indirectly, given that single species can be found under very diverse topographic conditions (*Dubay & Witt, 2014*). The remaining environmental variables were transformed *via* principal component analyses to understand relative variable importance using the function 'rda' in the R package *vegan* (*Oksanen et al., 2022*).

Data were further restricted to variables less affected by temperate latitude seasonality to better reflect potential differences within populations' breeding niches. That is, I retained the ENVIREM variables of annual potential evapotranspiration, Thornthwaite aridity index, climatic moisture index, Emberger's pluviometric quotient (a measure for differentiating Mediterranean climates), minimum temperature of the warmest month (generally corresponding with the breeding season for migratory taxa), potential

evapotranspiration of the driest quarter, potential evapotranspiration of the warmest quarter, potential evapotranspiration of the wettest quarter, and topographic wetness index (*Title & Bemmels, 2018*). I proceeded with all subsequent analyses using this subset of environmental data, and I report results related to this set of variables.

## Environmental comparisons

To assess different 'ecopopulations' (*i.e.*, populations as defined by unique environments) and to identify how well-partitioned *Progne* taxa are ecologically, I used the aforementioned individual environmental data layers corresponding to breeding niches to perform linear discriminant analyses in *R*, using the 'lda' function in the package *MASS* (*Venables & Ripley, 2002*). These tests partition individuals based on the environmental data, and suggest hypotheses for group assignments for individuals among known group assignments and presumed groups based on environmental characteristics (*Cooper et al., 2021*). I performed these tests for all *Progne* species and within the polytypic *P. subis*. I also verified whether the number of taxa recognized is supported by the environmental data by performing gap-statistic analyses of *k*-means clusters using the function 'fviz_nbclust' in the *R* package *factoextra* (*Kassambara & Mundt, 2020*).

## Niche modeling

For each species, I created ecological niche models using a presence-only method, minimum volume ellipsoids, following *Cooper et al. (2021)*, adapting an *R* script originally provided by J. Soberón (*Osorio-Olvera et al., 2020*). I opted for the use of minimum volume ellipsoids as an estimate of the fundamental niche, rather than use a more complex model to estimate the realized niche, as niche evolution pertains to the overall fundamental niche of these species (*Jiménez et al., 2019*). These minimum volume ellipsoids were created using the data layers corresponding to important variables for the breeding niches, as determined *via* aforementioned principal component analysis. I fit minimum volume ellipsoids with an inclusion level of 90% using the 'cov.mve' function in the *R* package *MASS* (*Venables & Ripley, 2002*). I used this inclusion level to exclude potential vagrant individuals while not removing too much information from range-restricted species. After calculating the MVE, I then created a raster of suitability based on the Mahalanobis distance of the environmental conditions of each raster cell to the calculated niche ellipsoid, opting for a Mahalanobis distance to account for the distance to a distribution of points and not to a single point (*Mahalanobis, 1936*; *Cooper et al., 2021*).

While not necessary for niche comparisons, I also computed binary presence-absence maps for each species from these niche models to observe the predicted distributions of each taxon. Given that the distances of occurrence points from the minimum volume ellipsoid are asymmetric with a heavy right skew, I used Gamma distributions fit to the environmental distance distribution of occurrences to create thresholds of 75%, 85%, 90%, 95%, and 99% data inclusion using the *R* function 'fitdist' in the package *fitdistrplus* (*Delignette-Muller & Dutang, 2015*). Binary maps created from the thresholded models appeared to be most accurate at modeling known distributions at 90–99% thresholds for localized species and for 75–90% thresholds for more widespread populations. Thresholded

maps included all areas accessible to the species, and thus overpredicted spatial distributions and projected occurrences into other areas where the taxa do not occur.

## Ecological niche comparisons

Within this system, I sought to evaluate whether species conform to the null hypothesis of niche conservatism or if taxa demonstrated evidence for ecological niche shifts through evolutionary time. To determine if ecological niches differ, I opted for pairwise comparisons between taxa's ecological niche models to obtain measurements of niche similarity following the pipelines of *Warren, Glor & Turelli (2008)* and *Cooper et al. (2021)*. These tests involve comparing the similarity between two species ecological niche models, calculating Schoener's $D$ statistic with the 'nicheOverlap' command in the *R* package *dismo* (*Hijmans et al., 2021*), to a null distribution Schoener's $D$ derived from comparing each species' ecological niche model to random models derived from the other species' **M** (*i.e.,* its accessible biogeographic area). Schoener's $D$ evaluates the similarity between rasters as a whole, such that identical rasters have $D = 1$ and wholly different rasters have $D = 0$ (*Schoener, 1968*; *Warren, Glor & Turelli, 2008*; *Warren, Glor & Turelli, 2010*; *Glor & Warren, 2011*). By creating a random model from points drawn from within the ecoregion of a given species, it is possible to ascertain whether the difference between the taxa of interest is significantly different with respect to the null comparisons. To ensure I had sufficient power for the comparisons, I created 100 random niche models within the **M** of each species, such that each true comparison was compared to the null distribution of each species individually compared to 100 null models from the other species' **M** (*Cooper et al., 2021*). When comparing the true value of the comparison to both null distributions, I used an $\alpha = 0.05$ treating it as a two tailed test, such that taxa with $p < 0.025$ with respect to both null distributions were considered have diverged ecologically. If only one test has $p < 0.025$, I made a note of these taxa as potentially in the process of diverging, but as not yet possessing differentiated niches. I made note of taxa that had niches more closely related than expected by chance ($p > 0.975$) to see which taxa most strongly conform to the null hypothesis of niche conservatism. Several different combinations of non-significance for both comparisons are possible using this system, which may be indicative of a 'spectrum' of ongoing niche diversification. I therefore scored comparisons as −2 ($p < 0.025$ for both distributions), −1 ($p < 0.025$ for one comparison), 0 (random variation), 1 ($p > 0.975$ for one comparison), and 2 ($p > 0.975$ for both comparisons).

## Historical niche reconstruction

To ascertain if any observed ecological niche shifts represent true evolutionary shifts, I created historical ecological niche reconstructions against which niches could be compared. Historical niche reconstructions were performed using the R packages *ellipsenm* (*Cobos et al., 2022*), *geiger* (*Pennell et al., 2014*), *nodiv* (*Borregaard et al., 2014*), and *nichevol* (*Cobos, Owens & Peterson, 2020*), individually looking at the variables identified as being important for the breeding season for *Progne* martins. Reconstructions utilized the aforementioned accessible areas (**Ms**) for each species to account for accessible and inaccessible environmental space for each taxon. This is a necessary control step to ensure

that species' historical reconstructions are not projecting or restricting species' ecological niches based on biogeographic artifacts resulting from the geographic areas in which species reside, thereby creating more accurate hypotheses regarding ancestral niches by allowing for uncertainty (*Saupe et al., 2018*). I used the same ecological characters as described above for the pairwise comparisons for consistency (*Cooper & Soberón, 2018*). I used a phylogenetic tree of species' relationships based on a UCE study of the family Hirundinidae provided by Clare E. Brown, Subir Shakya, and Fred Sheldon (*Brown, 2019*). This tree is missing two taxa due to poor DNA reads, namely Galápagos *Progne modesta* and Cuban *P. cryptoleuca* Martins. I added *P. cryptoleuca* to the tree as sister to Caribbean Martin *P. dominicensis* halfway between the node and the base of the dendrogram based on other phylogenetic information indicating that these taxa form a sister-pair (*Moyle et al., 2008*). I performed *nichevol* analyses of each ENVIREM character to assess how ecological niches shifted through time, and to identify which species experience the largest evolutionary shifts.

## RESULTS

### Environmental differences and clustering

The top explanatory variables for the first principal component were annual evapotranspiration (70.3%) and Emberger's pluviometric quotient (27.8%). For the second principal component, they were Emberger's pluviometric quotient (71.5%) and annual evapotranspiration (27.3%). I observed broad ecological niche overlap within the genus generally, with respect to individual environmental variables and in terms of environmental variables transformed by principal component analysis. Species that occupy different extreme habitats (*e.g.,* desert *vs.* rainforest) may show differentiation along individual environmental axes that reflect these differences, but few taxa showed overall differences in realized niches (Supplemental Material: 01_data_format: Analyzing Environmental Data). Using gap-statistic analysis, I determined the number of 'ecospecies' within the genus *Progne* to be 5 (Supplemental Material: 01_data_format: Human-based designation vs. machine-based designations). However, when classifying the full set of occurrence data into five groups using *k*-means, none of these ecospecies corresponded clearly to any described taxon, and no individual taxon is fully within any *k*-means group. Using discriminant function analyses, the most accurately reconstructed species from environmental data were *P. murphyi* (90% clustered together) and *P. subis* (94%). The greatest confusion was related to Gray-breasted Martin *P. chalybea*, a species that is wide-ranging both geographically and environmentally. Most individuals of Cuban Martin *P. cryptoleuca*, Caribbean Martin *P. dominicensis*, Sinaloa Martin *P. sinaloae*, and Brown-chested Martin *P. tapera*, were ascribed to the same group as *P. chalybea* (Table 2).

Within *P. subis*, I found that the most-supported number of ecopopulations in this clade is 1. This result holds true even when all taxa except nominate *P. s. subis* are compared. Despite this, discriminant function analyses of the three currently recognized subspecies of *P. subis* had a high level of success in discriminating taxa, with accuracies of 96% for *P. s. arboricola*, 96% for *P. s. hesperia*, and 99% for *P. s. subis* (Supplemental Materials; 01_data_format: Are Progne differentiating environmentally?).

**Table 2** **Population assignments for distribution points from each species based on environmental data.** Percent of individuals correctly clustered to original species using environmental data in a discriminant function analysis, with original (correct) taxon assignments being shown as rows and predicted taxon cluster being shown as columns, with accurate groupings falling along the bolded diagonal. Values shown are rounded percentages and thus may not exactly add to 100, and taxa are listed alphabetically.

| *Progne* species | chalybea | cryptoleuca | dominicensis | elegans | modesta | murphyi | sinaloae | subis | tapera |
|---|---|---|---|---|---|---|---|---|---|
| *chalybea* | 78 | 0 | 2 | 0 | 0 | 2 | 0 | 6 | 11 |
| *cryptoleuca* | 94 | 0 | 6 | 0 | 0 | 0 | 0 | 0 | 0 |
| *dominicensis* | 62 | 0 | 29 | 0 | 9 | 0 | 0 | 0 | 0 |
| *elegans* | 9 | 0 | 0 | 53 | 0 | 0 | 0 | 27 | 11 |
| *modesta* | 0 | 0 | 25 | 0 | 75 | 0 | 0 | 0 | 0 |
| *murphyi* | 10 | 0 | 0 | 0 | 0 | 90 | 0 | 0 | 0 |
| *sinaloae* | 80 | 0 | 0 | 0 | 0 | 0 | 10 | 10 | 0 |
| *subis* | 1 | 0 | 0 | 3 | 0 | 0 | 0 | 94 | 1 |
| *tapera* | 51 | 0 | 1 | 3 | 0 | 0 | 0 | 21 | 22 |

**Table 3** **Assignment of *Progne subis* individuals to population based on environmental data.** Percent of individuals correctly clustered to original subspecies within *P. subis* using environmental data in a discriminant function analysis, with original (correct) taxon assignments being shown as rows and predicted taxon cluster being shown as columns, with accurate groupings falling along the bolded diagonal. Values shown are rounded percentages and thus may not exactly add to 100, and taxa are listed alphabetically.

| *Progne subis* populations | arboricola (Interior) | arboricola (Pacific) | hesperia | subis | Mexican pops. |
|---|---|---|---|---|---|
| *arboricola* (Interior) | 93 | 1 | 1 | 0 | 0 |
| *arboricola* (Pacific) | 12 | 82 | 4 | 0 | 2 |
| *hesperia* | 4 | 1 | 90 | 0 | 4 |
| *subis* | 0 | 0 | 0 | 100 | 0 |
| Mexican pops. | 32 | 8 | 12 | 8 | 40 |

Subdividing these taxa further into the five groups based on region, habitat, and behavior still showed high support for each described subspecies, with accuracies of 93% for *P. s. arboricola*, 90% for *P. s. hesperia*, and 100% for *P. s. subis* (Table 3). Additionally, Pacific *P. s. arboricola* populations were largely differentiable from interior *arboricola* populations, with 82% of individuals being correctly identified and only 12% of individuals erroneously assigned as interior *arboricola*. Mexican populations were split between being considered their own entity (40%) and being considered an extension of interior *arboricola* (32%).

## Ecological niche divergence

Niches appeared to be largely conserved within the genus *Progne*, with a failure to reject the null hypothesis in 69% of pairwise tests; significant failures to reject with respect to both comparisons were only found in two pairwise comparisons: *P. subis vs.* Southern Martin *P. elegans* and *P. subis vs. P. tapera*) (Table 4A; Fig. 2). Only four pairwise tests showed definitive ecological niche divergence between test taxa: *P. chalybea vs. P. elegans, P. chalybea vs, P. murphyi*, *P. cryptoleuca vs. P. elegans*, and *P. cryptoleuca vs. P. murphyi*. Rejection of the null hypothesis for only one of the two pairwise comparisons was observed in six more comparisons, most involving combinations with *P. dominicensis*, *P. tapera*, *P.*

**Table 4  Environmental niche comparisons between different populations.** Significance of comparisons of ecological niches using random distributions drawn from each test species. Negative values indicate niche divergence, and positive values extreme conservatism, with only a score of '−2' being considered significant niche divergence. (A) Interspecific niche comparisons within Progne. (B) Intraspecific niche comparisons within *P. subis*.

| *Progne* taxa | chalybea | cryptoleuca | dominicensis | elegans | modesta | murphyi | sinaloae | subis | tapera |
|---|---|---|---|---|---|---|---|---|---|
| | | | | **(A)** | | | | | |
| chalybea | – | – | – | – | – | – | – | – | – |
| cryptoleuca | | – | – | – | – | – | – | – | – |
| dominicensis | | | – | – | – | – | – | – | – |
| elegans | −2 | −2 | -1 | – | – | – | – | – | – |
| modesta | | | | −1 | – | – | – | – | – |
| murphyi | −2 | −2 | −1 | 1 | | – | – | – | – |
| sinaloae | | | | | | | – | – | – |
| subis | | 1 | | 2 | 1 | 1 | | – | – |
| tapera | 1 | | −1 | | −1 | −1 | | 2 | – |

| *Progne subis* population | arboricola (Interior) | arboricola (Pacific) | hesperia | subis | Mexican pops. |
|---|---|---|---|---|---|
| | | | **(B)** | | |
| arboricola (Interior) | – | – | – | – | – |
| arboricola (Pacific) | 2 | – | – | – | – |
| hesperia | | 2 | – | – | – |
| subis | | 1 | 1 | – | – |
| Mexican pops. | −1 | 1 | −1 | | – |

*murphyi*, and *P. modesta* (Table 4A). Instances of divergence were not limited to widespread species or limited-range species, with one instance of divergence found between two wide-ranging species (*Progne chalybea* and *P. elegans*), two instances between widespread and restricted-range species (*P. chalybea* and *P. murphyi*, *P. elegans* and *P. cryptoleuca*), and one between two restricted-range species (*P. murphyi* and *P. cryptoleuca*). Conversely, ecological niche comparisons that were most similar (*i.e.,* the null of conservatism was the least likely to be rejected) involved the ecologically diverse *P. subis*.

Among subpopulations of *Progne subis*, no comparisons were able to conclusively reject the null hypothesis of niche conservatism (Table 4B). Only two pairwise comparisons rejected the null hypothesis for one of the two comparisons: *P. s. arboricola* (Rocky Mountains) *vs. P. s.* unknown (Mexico), and *P. s. hesperia vs. P. s.* unknown (Mexico). The most similar taxa appeared to be *P. s. arboricola* (Pacific Coast) and both *P. s. arboricola* (Rocky Mountain) and *P. s. hesperia*; however, it is unclear whether this similarity is informative or merely within the variation of the spectrum of what can be considered niche conservatism.

## Ecological niche reconstructions

Reconstructions were created for each variable independently. The ancestral *Progne* was found to have a broad ecological niche in most aspects, though (in some cases) not quite as broad as the most basal extant species, *P. tapera* (Fig. 3; see Appendix S1; Section C & D). Reconstructions consistently showed instances of niche contraction in geographically

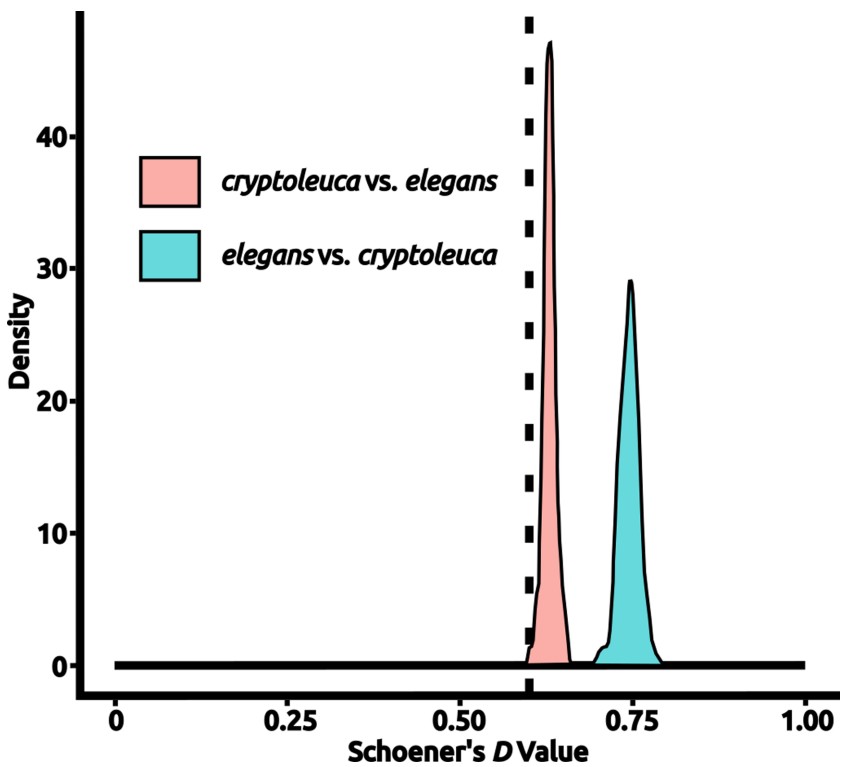

**Figure 2 An example test of niche equivalency.** Density histograms show the distribution of values drawn from calculating Schoener's *D* for niche models created from random points in each species' accessible area to the actual niche model created for the other species, with the test statistic being derived from a direct comparison of the "true" models created from each species' occurrences. Values of Schoener's *D* vary between 0 (completely different) and 1 (wholly identical). In this example, *Progne elegans* is compared to *P. cryptoleuca*, with a density histogram of values from random tests being shown; the test statistic is shown as a vertical dashed line. In this particular example, the taxa were found to have significantly different niches with respect to both *D* distributions ($P < 0.025$). Plots for all comparisons are available in Appendix S1. Figure prepared with Inkscape 1.2 (*Inkscape Project, 2022*).

localized taxa, especially among the restricted-range species such as *P. murphyi*, *P. sinaloae*, and *P. cryptoleuca*. Niche expansion was observed for several traits, such as Emberger's Pluviometric Quotient (designed to separate Mediterranean climates), but such expansions were mostly observed in widespread taxa or taxa that breed at high latitudes. Some taxa also experienced niche expansion with respect to their sister species, further indicating flux in the occupied environmental areas within the clade. Wide ranging taxa, especially *P. subis* and *P. chalybea*, demonstrate this niche expansion with respect to their most recent common ancestor with their less widespread sister taxa. Most niches, however, were similar to the broad ancestral niche, or were contained within the space of the broad ancestral niche. Unsurprisingly, some taxa, especially insular taxa, inhabit ecological niches that cannot be characterized completely owing to limited environmental conditions being present within the species' accessible area.

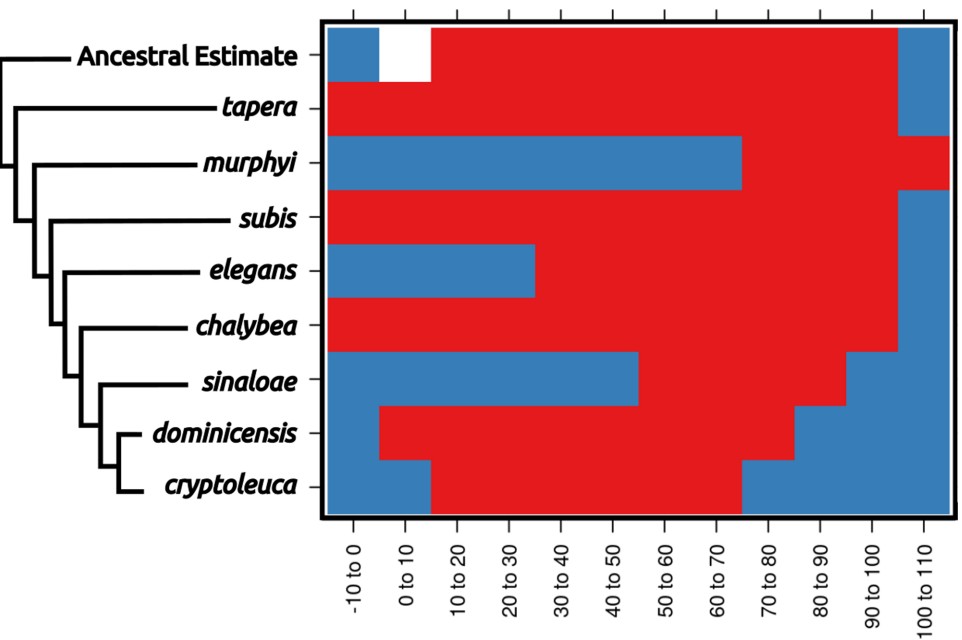

**Figure 3 Ecological niche reconstructions for aridity tolerance for the genus *Progne*.** Niche ranges are shown across the plot as occupied (red), unoccupied (blue), or uncharacterized (white). Note that the historic niche appears to be rather large and encompassed most of the niche now occupied by the entire genus. Widespread modern taxa, such as *P. tapera*, *P. subis*, and *P. chalybea*, have large niches similar to the ancestral state, while restricted range taxa have evolved into narrower niche spaces (with *P. murphyi* even showing evidence for niche expansion), and some of these taxa would appear to have niche divergence if not compared within the larger framework of the genus *Progne*. A scaleless phylogeny based on *Brown (2019)* is shown for reference, with the basal node shown as the "Ancestral Estimate". Figure prepared with Inkscape 1.2 (*Inkscape Project, 2022*) and R package *rasterVis* (*Perpiñán & Hijmans, 2023*).

## DISCUSSION

I found widespread support for niche conservatism within *Progne*. Historical reconstructions demonstrate that members of *Progne* are largely overlapping in ecological space both with their congeners and the presumed niche of their ancestral taxon. Thus, current niche similarities are not the result of ecological niche convergence, and observed divergences appear to be the result of niche partitioning wherein species niches contract with respect to their ancestral state. The most similar ecological niches were found among basal martins with broad distributions, namely between *P. subis* and both *P. tapera* and *P. elegans*.

I found only a few instances of ecological niche divergence overall, with these divergences being between non-sister taxa. Two of the incidences of niche divergence were between *P. elegans*, the outgroup of the 'white-breasted' martin clade, and species found in the same clade (*P. chalybea* and *P. cryptoleuca*). This group includes multiple morphologically conservative species that are not always identifiable in the field, but that possess largely discrete breeding ranges in diverse habitats, ranging from mid-elevation montane forests

to coastal scrub, palms, and urban areas (*Fang & Schulenberg, 2020*; *García-Lau & Turner, 2021*; *Perlut & Williams, 2021*).

## Ecological niche conservatism in Progne

I was unable to reject the null hypothesis of niche conservatism for most of *Progne*, providing further evidence that organisms are unlikely to have rapid niche shifts while diversifying (*Peterson, Soberón & Sanchez-Cordero, 1999*). Evolutionary reconstructions indicate that the ancestral *Progne* was a South American species with a broad ecological niche, and that it was at least partially migratory (*Brown, 2019*), similar to the ecology of the extant *Progne tapera*. The next basal groups included *P. subis*, a long-distance North American migrant, and the range-restricted species pair *P. murphyi* and *P. modesta*. It appears that *P. subis* is descended from a South American species (*Brown, 2019*), perhaps similar to the recent North-to-South Hemisphere colonization of Barn Swallows *Hirundo rustica* in Argentina (*Martínez, 1983*). Similar north-to-south shifts and geographic isolation appear to have led to diversification within the 'white-breasted' martins as well, resulting in three allopatric species breeding across the North American and Caribbean tropics (*Brown, 2019*). Such events demonstrate the importance in geographic isolation for driving diversification within *Progne* and support the idea of *Progne* undergoing a 'geographic radiation' of speciating while maintaining similar ecological niches among the descendant species (*Peterson, Soberón & Sanchez-Cordero, 1999*; *Simões et al., 2016*).

Ecological niche differences exist among a few extant *Progne*, but these appear to be the result of ecological niche partitioning and not true ecological niche shifts. As species have colonized isolated regions and have continued to adapt regionally, they have experienced niche contractions, possibly as a result of specializing to conditions in these regions, or moderate niche shifts, wherein they have expanded their ecological tolerance within their specific geographic distributions. The former is demonstrated well by *Progne murphyi*, a species restricted to the coast of Peru. The restricted ecological niche of *P. murphyi* is reminiscent of taxon cycles in the Caribbean: species diversify, and descendant species become more restricted (and sometimes more ecologically specialized) through time (*Ricklefs & Cox, 1972*; *Engler et al., 2021*). Results within the genus *Progne* indicate that tests of ecological niche divergence should try to account for ancestral niche states to understand whether observed niche evolution is novel or reflective of a different evolutionary process, such as partitioning of the ancestral state.

*Progne subis* appears to be a microcosm of these phenomena, with the species as a whole occupying a broad ecological niche, with individual populations evolving for specific environmental conditions within the overall niche space. Within *P. subis*, evidence for niche divergence is lacking, with results reflecting a continuum of variation between descendant populations partitioning environments than any true ecological shift. In southwestern North America, cactus-nesting *P. s. hesperia* are found in close geographic proximity with montane *P. s. arboricola,* yet I found ecological niche overlap between these taxa that differ drastically with respect to habitat preference. These populations have niches within the broader *Progne* subis niche reminiscent of other extant species occupying subsets of the broader *Progne* niche elsewhere (*e.g., P. murphyi* and *P. sinaloae*), further supporting
the notion of niche partitioning and niche specialization across a geographic mosaic, as opposed to significant ecological niche evolution leading to large ecological shifts.

Diversification *via* niche partitioning of a broader ancestral state may be more common than is realized in continental taxa, given the propensity of groups like *Zosterops* to undergo taxon cycles in montane regions (*Ricklefs & Cox, 1972*; *Melo, Warren & Jones, 2011*; *Pearson & Turner, 2017*; *Engler et al., 2021*). Likewise, research is demonstrating the propensity for related populations to partition and specialize on different food sources across space and time (*Benkman et al., 2009*; *Cenzer, 2016*; *Alonso et al., 2020*), illustrating ways in which niches can be partitioned locally to allow for co-occurrence at broad spatial scales or to allow for regional diversification. Other complexes also appear to demonstrate niche partitioning, in part maintained by competition, including chipmunks (*Neotamias*) in the intermontane west of North America and within African *Turdus* that have diversified broadly into montane and lowland forms across the continent (*Bowie et al., 2005*; *Kelt et al., 2023*). *Neotamias* in particular show features of niche partitioning in that the removal of one species allows other species to expand to occupy larger ecological niches, with individual species demonstrating ecological specialization within the larger *Neotamias* niche (*Chappell, 1978*). Further research into geographic radiations, especially those that occur mixed allopatry and parapatry in montane regions, will shed more light on the patterns and processes of ecological niche partitioning.

## Assessments of niche in geographically widespread groups

Assessments of ecological niches are still frequently performed across political boundaries or study area boundaries, and thus do not account for the bias introduced by including sites and associated environments not accessible to the study taxa over relevant time periods (*Soberón & Peterson, 2005*; *Barve et al., 2011*; *Owens et al., 2013*; *Peterson & Anamza, 2015*; *Cooper & Soberón, 2018*; *Song et al., 2020*). Similarly, models that focus explicitly on the realized niche, while useful for conservation, may not account for the broader fundamental niche of a species that may be accounted for in broader analyses (such as those employed here) (*Jiménez et al., 2019*). These methods can bias niche estimates and therefore bias distribution models in current environments. These issues can also compound when comparing ecological niches through time to understand their evolution (*Saupe et al., 2018*).

Another less discussed oversight, however, is that of population-level variation within species. What constitutes a species can be contentious (*Watson, 2005*), so what is recognized as a species in the literature can vary greatly between taxonomic authorities (*Barrowclough et al., 2016*; *Garnett & Christidis, 2017*; *Raposo et al., 2017*). The effects of these oversights are easily missed in studies focusing on 'species-level' diversification. I focused on sets of populations of *P. subis*, and results illustrated the processes that are occurring on a more unitary, fine-scale basis within this one species.

## CONCLUSIONS AND FUTURE DIRECTIONS

Future research on *Progne* should focus on understanding relationships between populations across ranges of species (particularly *P. subis*) and in clarifying breeding

and non-breeding distributions within the genus. Several species and populations are poorly known, most notably populations of *P. subis* and *P. sinaloae* in western Mexico. Ecological analyses such as those developed here are useful for helping identify ecological specialization between closely related taxa, even when those taxa occupy portions of the broad ancestral niche and can guide future efforts to plan sampling for phylogeographic analyses. Ecological niche analyses can also focus more on the temporality of ecological niches, specifically considering nest sites, wintering sites, and the seasonal occupancy of each species or population, to understand environmental conditions necessary for species survival and to understand how niches are or are not conserved through annual cycles (*Nakazawa et al., 2004*; *Peterson et al., 2005*).

Ecological niche divergence is not a driver of or an inevitable consequence of diversification, and geographically widespread radiations can exhibit niche conservatism. *Progne* martins provide an excellent case study of cryptic diversification driven by habitat specialization, behavioral differences, and allopatry. Species pairs that occupy small, specialized ranges or different latitudinal areas can show ecological niche differences from related taxa while still occupying portions of the ancestral ecological niche. Within the ecologically diverse *P. subis*, niche conservatism between described subspecies appears pervasive, notwithstanding divergence in habitat preferences and behavior. These results highlight how niche divergence can be decoupled from diversification and highlight the need for extensive geographic sampling when studying gene flow and variation within taxa.

## ACKNOWLEDGEMENTS

I would like to dedicate this article to Richard G. Levad, whose personal mentorship helped inspire me to study birds and whose work shed light on Colorado's most enigmatic birds, including *Progne subis* populations there. I would like to thank Kim Potter, Jason Beason, Glenn P. Giroir, and Carolyn Gunn for further instilling in me an interest in *Progne* martins. A. Townsend Peterson provided critical comments on this manuscript. Clare E. Brown provided comments regarding the project and enabled this research with her graduate work. Additional phylogenetic assistance was provided by Subir Shakya and Fred Sheldon. Coding advice and assistance were provided by Marlon Cobos, Fernando Machado-Stredel, Jorge Soberón, and Joseph D. Manthey.

### Funding
This research was funded by the Institutional Research and Academic Career Development Award (IRACDA; NIH 2K12GM063651) to the University of Kansas. The funders had no role in study design, data collection and analysis, decision to publish, or preparation of the manuscript.

### Grant Disclosures
The following grant information was disclosed by the author:

Institutional Research and Academic Career Development Award: NIH 2K12GM063651.

## Competing Interests

The author declares that there are no competing interests.

## Author Contributions

- Jacob C. Cooper conceived and designed the experiments, performed the experiments, analyzed the data, prepared figures and/or tables, authored or reviewed drafts of the article, and approved the final draft.

## Data Availability

The code is available at GitHub and Zenodo:

– https://github.com/jacobccooper/progne_niche_evolution.

- *Cooper (2024)*. Code and data for: Ecological niche divergence or ecological niche partitioning in a widespread Neotropical bird lineage. Zenodo. https://doi.org/10.5281/zenodo.10778167.

The data are available from the ENVIREM dataset (Title & Bemmels, 2018): https://envirem.github.io/.

The occurrence data are available from GBIF: https://doi.org/10.15468/dl.btsx3g.

The phylogenetic information was supplied by Clare E. Brown, Subir Shakya, and Fred Sheldon and is based on *Brown (2019)*.

## Supplemental Information

Supplemental information for this article can be found online at http://dx.doi.org/10.7717/peerj.17345#supplemental-information.

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
