# Peer review of "Ecological niche divergence or ecological niche partitioning in a widespread Neotropical bird lineage"

_PeerJ, doi:10.7717/peerj.17345_

## Round 0.1 · original submission · Major Revisions

Please ensure all reviewer suggestions are addressed as your paper will be sent to a second round of reviews. I look forward to receiving the revised manuscript.

Reviewer 1 ·

Basic reporting

In my opinion, this is a very good and interesting study. It is quite clear, very well written and the English is 100%. This study cited recent and important papers in the field of study. Thus, the context/background is appropriate. Sections of the manuscript are well structured and present, mostly.
Results are interesting.


Please find below some additional comments and suggestions on specific sections of the manuscript:

Abstract
The Abstract starts well and properly explains the objective of the study. But the author jumps directly to the results. As the word limit is 500, and you have 273, the author could add some sentences explaining the methods used in your study. Results and conclusions are fine.

Introduction
The first sentence is a bit strange – “ecological niche” appears twice. Please review it.
Paragraphs 1 and 2 run well with the presentation of major facts involved in the research topic, with the citation of recent papers, and some important traditional ones.
In Paragraph 3, the author properly cites and explains the three major scenarios for niche evolution, but cites Figure 1. I consider it a bit strange (calling the Results in the Introduction…). Maybe, the author could re-write/mix some sentences to present the three scenarios and provide the examples already cited, without calling Figure 1. But this might be a concern of the Editor. If allowed, just keep it, as it is well written.
Paragraph 4. Again, I consider strange calling Table 1. Actually, it is not necessary here. The first sentence is enough by itself. Lines 80-82 – these references are very old. No recent? Maybe, the author can add the review by Winkler et al 2020, from Birds of the World https://birdsoftheworld.org/bow/species/hirund2/cur/species#genusProgne, or others specific to the species mentioned on the lines below. Anywhere, in the beginning of this paragraph, mention the existence of 9 species of Progne.
Paragraph 5 is fine.
Last paragraph. The ideas and reasons are very clear and run well, but the author has not clearly presented the objective of this study (it is “cryptic”). Please think about.
Overall, the Introduction is very well written, and its reading runs well, with appropriate moving between consecutive paragraphs.

Methods
General Data – It is very well written and clear.
Line 122. Delete the space before the comma.
Line 124. The reference is in italics.
The methods sound as appropriate.
Environmental data
Lines 177-182. This part appears to be Results. Should it really be here in the Methods ?
Sentence of the first principal component: it would be better to divide it in two sentences, or rewrite. Annual evapotranspiration and thermicity explained nearly nothing.
Next paragraph. Again, it appears that you have results in the Methods.
Line 207. The reference is in italics.
Other sections of the Methods are fine and very well explained.

Results
Results were properly divided in three sections.
The writing is quite clear and well organized.
Tables and figures were called in adequate places.
I found nothing to fix. Results are very good and interesting.

Discussion
In general, the Discussion (initial section and 4.1) deals well with the Progne taxa. However, I consider that the Discussion could call a bit more examples of similar studies with other animals, as along lines 397-402.
Please check again if you discussed all aspects of your Results.
Some scientific names are not in italics.

Conclusions and future directions
This section is well though and present. I liked it.

References
I did not check if all cited papers were included here in the References section, nor vice-versa.
Kennedy. What is the problem with the volume and pages ?
Nakazawa. Words are with initials in capital.

Experimental design

All the concerns shown on the wright side of this space were properly achivied. See the section above for specific parts.
In general, the study was wery well planned, and used appropriate methods.
Results are convincing.

Validity of the findings

Yes, the Results are very good. Very well written, divided in appropriate sections, and placed in good tables and figures. The supplemental material it fully provided.
The conclusion is well though and presented.

Additional comments

Congratulations on this study. It is of high quality.

Reviewer 2 ·

Basic reporting

In this work, the author tested the null hypothesis of niche conservatism among the species of the genus Progne and explored how niche divergence can be decoupled from diversification. The author proposes to make the analyses at an interspecific level (between different species in the complex) and an intraspecific level (within subpopulations of one species in the genus), which provides a rounded view of distinct niche divergence scenarios.
Except for the Methods section, the manuscript is clear and well-written. The Introduction provides relevant information to understand the background and general objectives of the study. In the next section, I will comment on specific aspects of the Methods section that need revision and improvement.
The figures and tables are well done, and they either help to understand the context of the study or show relevant results.

Experimental design

Line 119: maybe a better name for this section is something like ‘Occurrence Data’ since the data cleaning described here only applies to occurrence records.
Lines 135-138 & 248-249: when you described how the Ms were created you mention that they were only used to identify potential error and to assign breeding records to the correct population. However, the Ms are used with a different purpose later in the manuscript. Moreover, in Section 2.5, it is not clear why the Ms are needed in the niche reconstruction process, or how these regions are used. Please provide more details on how the Ms are used in the historical niche reconstructions.
Line 139: is there any specific reason why you are using a threshold of 20 km to thin the data? This looks like a random decision.
Lines 193-195: even though it is mention here that, depending on the type of analyses performed, you used either PCs or raw variables, it be helpful to indicate in each of the following sections whether you are using one type of variable or the other. According to this lines, raw variables were used to fit the niche models and the first two PCs were used to perform niche comparisons; however, it is not clear how did you transform the niche model outputs into the different components needed to calculate Schoener’s D scores using PCs. Please add some lines where you clarify this point.
In my experience, when calculating niche similarity measures using models with a small number of climatic variables (PC1 and PC2 in your case), these scores are usually large, leading to the conclusion that the niches are not different, whereas if the niche models were fitted with more variables, it is more likely to observe niche differentiation. This is, the conclusions regarding the hypothesis test of niche conservatism may be sensitive to the dimensionality of the niche space. Have you consider repeating these analyses by including one or a few variables in the niche model to then calculate and compare the resulting Shoener’s D scores to see if the results change?
Lines 210-212: is there any specific reason why you are using minimum volume ellipsoids? Is there a rationale behind using a 90% inclusion level?
Lines 213-222: there is a lack of connection between the lines explaining how the niche models where fit and the lines explaining how to transform the continuous model outputs into categorical/discrete outputs. Why do we need to calculate Mahalanobis distances and how are these connected to the minimum volume ellipse estimated with the ‘cov.mve’ function? Please explain further.
You mention that the Mahalanobis distances obtained with the data are non-normal, however, it is not clear why we need to model these distances. Moreover, why are you proposing to model these distances with a Gaussian distribution? In lines 217-218, you mention that these distances are right-skewed but the Gaussian distribution is symmetric. What is more confusing is that, in your R code, you have the following line:
fitdist(ext2, dist=”gamma”, method=”mle”)
which means that you are fitting a Gamma distribution to the Mahalanobis distances through a maximum likelihood estimation approach. Thus, this is inconsistent with the manuscript. Besides the Gamma distribution, there are other possible distributions that can be used to model distances (e.g., a half-normal distribution), so you need to provide a rationale behind your model choice.
Additionally, line 220 is confusing: ‘models were thresholded to the following inclusion levels’. Are you modifying the inclusion levels in the ‘cov.mve’ function to fit different ellipses? Or, are these confidence levels of the probability distribution used to model the Mahalanobis distances? Moreover, why do you need to get thresholded output models for all these levels?
Lines 223-243: I found this paragraph misleading and it lacks important details. It is true that Schoever’s D scores and the test for niche conservatism are widely used in this field, but I’d like to invite you to add relevant details that the reader may need to be able to reproduce these analyses and that are often not included. For example, what are the objects or numbers that we need to calculate Schoener’s D and how do we get them from the niche models?
Lines 227-229 are particularly confusing; I suggest to rephrase it. Also, are 100 replicates enough to get a good estimation of the sampling distribution of the test statistic?
When explaining how the niche conservatism hypothesis was tested, the paragraph should contain the following elements (maybe even in this order to ensure clarity): explicitly state the null hypothesis, what is the test statistic and how is it calculated, how is the sampling distribution estimated, what is the criteria to reject or not the null hypothesis.
When explaining the criteria to reject or not the null hypothesis, the criteria is based on a confidence interval: ‘test above the 97.5% confidence interval’, ‘statistic fell below the 2.5% confidence interval’. There’s a lot of ambiguity in these sentences. One sentence is stated in terms of a ‘test’ and the other one is stated in terms of a ‘statistic’, I think you mean ‘test statistic’ in both sentences. This is correctly explained in Figure 2 though. Are you calculating two confidence intervals? I think that what you calculated is either a p-value or two values that determine the rejection region?
Lines 264-266: Can this be solved by setting a ‘seed’ number in your code so the analyses are reproducible?

Validity of the findings

The results, discussion, and conclusions need to be substantially modified after considering the comments related to the experimental design. I cannot satisfactorily evaluate the validity of all the findings given the lack of details and the questions I raised regarding some methodological aspects.

Additional comments

The ‘Supplemental Materials’ are heavily referenced in the manuscript and they consist of a series of R scripts and markdown files. Given the length of these files, it is not clear what specific file the reader should look at. I suggest pointing out specific sections of these materials so the reader is able to follow up on the text and reproduce the analyses. Additionally, in line 273, these materials are referred to as ‘Supplementary Materials’.
In the Appendix, there are some figures related to the niche reconstruction in which the names on either the x-axis or the y-axis (or both) are indistinguishable. Please modify accordingly and consider resizing the figures to shrink this overly long file.
In Tables 2, 3, and 4, consider writing the column names in a vertical fashion to make them more compact.
Lines 223-242: there are 5 citations to a paper led by the author, to me this looks excessive. The beginning of the paragraph makes it explicit that all the analyses described in this paragraph were done following that study, so this may be enough. Consider leaving only the first instance of this citation, which I consider important.
Lines 259-260: the second part of the sentence is repeated.
Line 271: ‘principal component analysis’ instead of ‘principal components analysis’.
Line 301: erase extra parenthesis.
In Table 4, you did not explain in the main text how to get and how to interpret the values presented in this table. Please add some lines in the main text so this is clear.

---

## Round 0.2 · Minor Revisions

Reviewer 2 has a few minor comments that can improve your article.

Reviewer 1 ·

Basic reporting

I think that the author provided convincing responses to my comments and suggestions, and the changes in the manuscript were appropriate.

The writting and the English are quite good.

The Introduction is very well written, and with the most pertinent literature.
The objectives have been improved.
the methods are appropriate and well explained.
Results are interesting and supported by good figures and tables.
The discussion is well done and calls an international audience.

That will be a great paper.

Experimental design

No comment.

Validity of the findings

No comment.

Additional comments

Congratulations on your study.
I consider it is a great contribution to this reaserch field.

Reviewer 2 ·

Basic reporting

This version of the manuscript is very well-structured and well-written.
The author addressed all the concerns that I raised in my previous review and I have no further major comments.

Experimental design

The author clearly presented the research questions and methods. He focused on testing whether the species within the avian genus of martins shows niche divergence. The methods that he used are appropriate and the Methods section contains enough information to follow up the analyses and interpret the results.

Validity of the findings

The discussion and conclusions are linked to the original research questions and presented in a concise and interesting narrative.

Additional comments

I only have a few minor comments, which do not impact the validity of the findings and the conclusions:

Line 272: erase 'In order' at the beginning of the sentence.

Lines 292-298: I think this information is related to the information included in section 2.4. If this is the case, I suggest to integrate this paragraph into section 2.4

Lines 300-302: Instead of creating a new section for this note, I suggest either moving this sentence to the Data Availability section or copying and pasting it into each subsection of the Methods where it applies.

Line 282: remove the extra dot before the citation.

---

## Round 0.3 · accepted · Accept

All requested changes were made, and the manuscript is ready for publication. Congratulations!